# Maternal occupational risk factors and preterm birth: Protocol for a systematic review and meta-analysis

**Haimanot Abebe Adane**[1]*, **Ross Iles**[1], **Jacqueline A. Boyle**[2], **Alex Collie**[1]

**1** Healthy Working Lives Research Group, School of Public Health and Preventive Medicine, Monash University, Melbourne, Victoria, Australia, **2** Monash Center for Health Research and Implementation, School of Public Health and Preventive Medicine, Monash University, Melbourne, Victoria, Australia

* haimanot.adane@monash.edu

**Data Availability Statement:** All relevant data are within the manuscript and its Supporting Information files.

## Abstract

### Introduction

Preterm birth, which accounts for 33.1% of neonatal death globally, is the main cause of under-five mortality. A growing number of studies indicate that occupational risk factors during pregnancy are linked to an increased likelihood of poor pregnancy outcomes. The effect of physical occupational risks on preterm birth has received very little attention, and previous reviews have produced inconclusive results. This systematic review aims to update the evidence on the relationship between maternal physical occupational risks and preterm birth.

### Method and analysis

We will search electronic databases including Ovid Medline, Embase, Emcare, CINAHL, Scopus, and Web of science to find peer-reviewed studies examining the relationship between six common maternal physical occupational risks (heavy lifting, prolonged standing, heavy physical exertion, long working hours, shift work, and whole-body vibrations) and preterm birth. Articles published in English after 1 January 2000 will be included without geographic restrictions. Two reviewers will screen titles and abstracts independently, and then select full-text articles that meet inclusion criteria. Methodological quality of the included studies will be evaluated using the Joanna Briggs Institute (JBI) critical appraisal method. The quality of evidence across each exposure and the outcome of interest will be examined by using the GRADE (Grade of Recommendations, Assessment, Development, Evaluation) method. Accordingly, a high level of evidence will lead to "strong recommendations". A moderate level of evidence will lead to "practice considerations". For all evidence levels below moderate, the message will be "not enough evidence from the scientific literature to guide policymakers, clinicians, and patients. If data permits, a meta-analysis will be conducted using Stata Software. In case where meta-analysis is not possible, we will perform a formal narrative synthesis.

### Discussion and conclusion

Evidence suggests that preterm birth is linked to a number of maternal occupational risk factors. This systematic review will update, compile, and critically review the evidence on the

**Funding:** The author(s) received no specific funding for this work.

**Competing interests:** The authors have declared that no competing interests exist.

**Abbreviations:** CI, Confidence Interval; MeSH, Medical Subject Heading; GRADE, Grade of Recommendations, Assessment, Development, Evaluation; OR, Odd Ratio; PICO, Population, Intervention, Comparison, Outcome; PRISMA, Preferred Reporting Items for Systematic Review and Meta-Analysis; PROSPERO, Prospective register of systematic reviews; PTB, Preterm Birth.

effect of maternal physical occupational risk on preterm birth. This systematic review will provide guidance to support decision-makers including maternal and child health services, other health care providers, and government policy agencies.

## Trial registration

**PROSPERO registration number:** CRD42022357045.

## Introduction

The number of pregnant women working has increased globally in recent decades [1]. In 2020, two-thirds of women of working age participate in the workforce (66.8%) in the European Union [2]. The Sixth European Working Conditions Survey showed that more than two-fifths (43%) of women worked physically demanding jobs, including those requiring them to adopt painful postures, 21% of women worked shift work, more than 15% worked more than 41 hours per week, and 14% worked night shifts [2].

In the authors home nation of Australia, over 61% of women are employed and 76.2% of women work throughout their reproductive age [3], a pattern replicated in most high-income countries. These and other recent studies show that many pregnant women engage in physically demanding work, long working hours, or shift work, and may be exposed to other occupational risk factors such as whole-body vibration [1, 4, 5]. The increasing exposure of reproductive-age women to workplace occupational risks has created concerns about the possible impact on maternal and neonatal health [6, 7].

Previous studies have shown that women who worked during pregnancy were at increased risk for poor maternal and newborn health, including preterm birth [8–12], defined as birth of the fetus before 37 weeks of pregnancy [13]. Rates of preterm birth range from 5% to 18% across 184 nations [14]. An estimated 15 million preterm births occur worldwide each year, with 1.1 million infant deaths' as a result of preterm birth, making it one of the leading causes mortality in children under 5-years of age [15]. In the authors' home country of Australia, in 2019 8.6% of newborn babies were born preterm [16].

Preterm birth also increases risk for noncommunicable diseases such as diabetes, hypertension, chronic lung disease, and heart disease later in adulthood [17–21]. The majority of preterm births happen spontaneously, however up to 30% are provider initiated [22]. Preterm birth is usually associated with factors such as multiple pregnancies, intrauterine infections, and chronic illnesses [23, 24].

Prior research has also identified occupational predictors of preterm birth including physical workload, prolonged standing, heavy lifting, long working hours, shift work, and whole-body vibration [25–30]. A systematic review of women who engaged in paid employment identified a positive association between long working hours and preterm birth. However, this review identified a significant association for long working hours, but not for shift work and the authors could not draw a definitive conclusion [31]. Another recent review examined the association between several other occupational risk factors and preterm birth. Pregnant women who work long hours in standing positions, perform heavy lifting or work shift-work or night shifts were found to be at increased risk of preterm birth [25, 26].

While the evidence from these reviews is useful, their authors have reporting conflicting or weak evidence and as such have concluded that it is challenging to provide explicit recommendations for clinical practice or policy [25, 31, 32]. A number of prior reviews have not utilised

rigorous methodological standards for reporting on study quality [30, 31]. None examined the impacts of whole-body vibration on preterm birth, and nor have they sought to differentiate between medically indicated or spontaneous preterm birth [25, 26, 28].

Furthermore, the included evidence in most reviews reflects working conditions of the 1960's and 2000's [12, 25]. In many occupations and nation, working conditions have changed dramatically throughout the early 21st century and thus the nature, prevalence and impacts of occupational physical health risks has also changed [33, 34]. In this review occupational physical exposure (as opposed to chemical or biological exposures or psychosocial exposures) are chosen as our primary emphasis because they are more commonly experienced by pregnant women [35], have been demonstrated to have a greater impact on women workers' sexual and reproductive health [34], because less attention has been given to physical factors than chemical and biological factors [12], and because many physical risk factors are modifiable with job re-design and thus there is potential for prevention.

Appraising the association between maternal occupational risks and preterm birth will provide beneficial knowledge for the obstetric community, occupational health services and policies. The aim of this systematic review is to assess the effect of maternal physical occupational risks (defined as including prolonged standing, heavy lifting, heavy physical demanding work, shiftwork, long working hours and occupational whole-body vibration) on preterm birth. Publishing this review protocol will benefit to reduce the impact of the review authors biases, increase the transparency of the methods and processes, reduce the possibility of duplication, and allow peer review of the planned methods.

## Methods and materials

This systematic review protocol follows the guideline recommended in Preferred Reporting Items for Systematic Review and Meta-Analysis (PRISMA-P) [36] and has been registered in PROSPERO database (International Prospective Registerer of Systematic Reviews) with registration number CRD42022357045.

### Eligibility criteria

**Inclusion criteria.** Articles that meet the following criteria will be included in this systematic review.

**Study population.** Singleton pregnant women who engaged in paid work during pregnancy (both nulliparous and multiparous women).

**Study design.** Observational research studies including prospective cohort, retrospective cohort, case control and cross-sectional studies, and quantitative interventional studies that examined the effect of maternal physical occupational risk on preterm birth.

**Comparator(s)/control.** Pregnant women who engaged in paid work during pregnancy and had less or no exposure to the maternal physical occupational risks.

**Outcome.** Preterm birth: babies born alive less than 37 weeks of pregnancy. If data permits we will also examine different aspects of preterm birth including: Extremely preterm (<28 completed weeks of gestation), Very preterm (28 - <32 weeks completed weeks of gestation), Moderate preterm (32 - <34 completed weeks of gestation), and Late preterm birth (34 - <37 completed weeks of gestation).

*Exposures.* One or more of the most common physical occupational risks including heavy lifting, prolonged standing, heavy physical workload, long working hours, shift-work, and whole-body vibrations. Due to a lack of standardised exposure definition, we will adopt a broad inclusion to ensure that we capture all articles reporting relevant exposure.

**Exclusion criteria.** The following studies will be excluded: (1) Non-primary studies: Reviews, case series, case reports, qualitative studies, editorials, commentaries, conference abstracts, and unpublished manuscripts; (2) Studies that examine preterm birth in women not working or that do not report employment status; (3) Studies published in languages other than English or before the year 2000; (4) Studies that investigate the effect of maternal occupational exposures other than those specified in the inclusion criteria, for example biological, chemical, or psychosocial occupational risks.

## Data sources and search strategy

We will conduct a systematic search of the following electronic databases: Ovid MEDLINE, Embase via Ovid, Ovid Emcare, Scopus, Web of Science, and CINAHL. The search strategy will be developed and carried out with the guidance of a librarian using a broad range of potential search and Medical Subject Heading (MeSH) and keywords for population, exposure and outcome. Articles published in English after January 1, 2000, will be included without geographic restrictions. Additionally, reference chaining will be conducted using relevant references that have been chosen for full-text examination. Only studies conducted on human populations and published in peer-reviewed journals will be included. A systematic search strategy combining exposure and outcome terms has been designed (See Table 1 and Table in "S2 File").

## Study selection process and software

All identified articles from the different databases will be imported into an EndNote library [37], duplicate articles will be removed and remaining studies will be imported into Covidence software [38]. Two independent reviewers will use Covidence to screen titles and abstracts. The full-text of those papers meeting study inclusion criteria will be downloaded and further screened by two independent reviewers to determine eligibility. If there is disagreement regarding eligibility between the two reviewers at either abstract, or full-text screening stage, a discussion will be held between the two reviewers to determine if consensus can be reached. A third reviewer will be consulted to determine study eligibility if consensus cannot be reached.

**Table 1. Search criteria.**

| Search term | |
|---|---|
| Maternal physical occupational risks AND | **Medical Subject Heading (MeSH):**<br>Occupational exposure OR lifting OR workload OR standing position OR employment OR shift work schedule OR work schedule tolerance OR |
| | "Occupational Exposure*".mp. OR (occupational activit* OR workload).mp. OR ((activ* adj2 (intens* or physical* or vigor*)) and (work* or occupation*)).mp. OR ((activ* adj2 (intens* or physical* or vigor*)) and (work* or occupation*)).mp. OR (heavy exertion* or heavy lifting or heavy work*).mp. OR (((demand* or drain* or exhaust* or fatigu*) adj2 physical*) and (work* or occupation*)).mp. OR (prolonged adj5 (standing or walking or bending or upright)).mp. OR (long work* hour* or long work*day*).mp. OR working time.mp. OR (shift work or afternoon shift or evening shift or morning shift or night shift or rotating shift or shift schedule).mp. OR "whole-body vibration".mp. |
| Preterm birth | **Medical Subject Heading (MeSH):**<br>Premature birth OR Infant premature OR infant extremely premature OR obstetric labour premature OR pregnancy outcome OR pregnancy complication OR |
| | **Keywords:**<br>((preterm or pre-term or premature) adj (birth* or infan* or babby or babies or neonat* or labo?r or deliver*)).mp. OR extreme* prematur*.mp. OR ("adverse pregnancy outcome" or "adverse birth outcome").mp. |

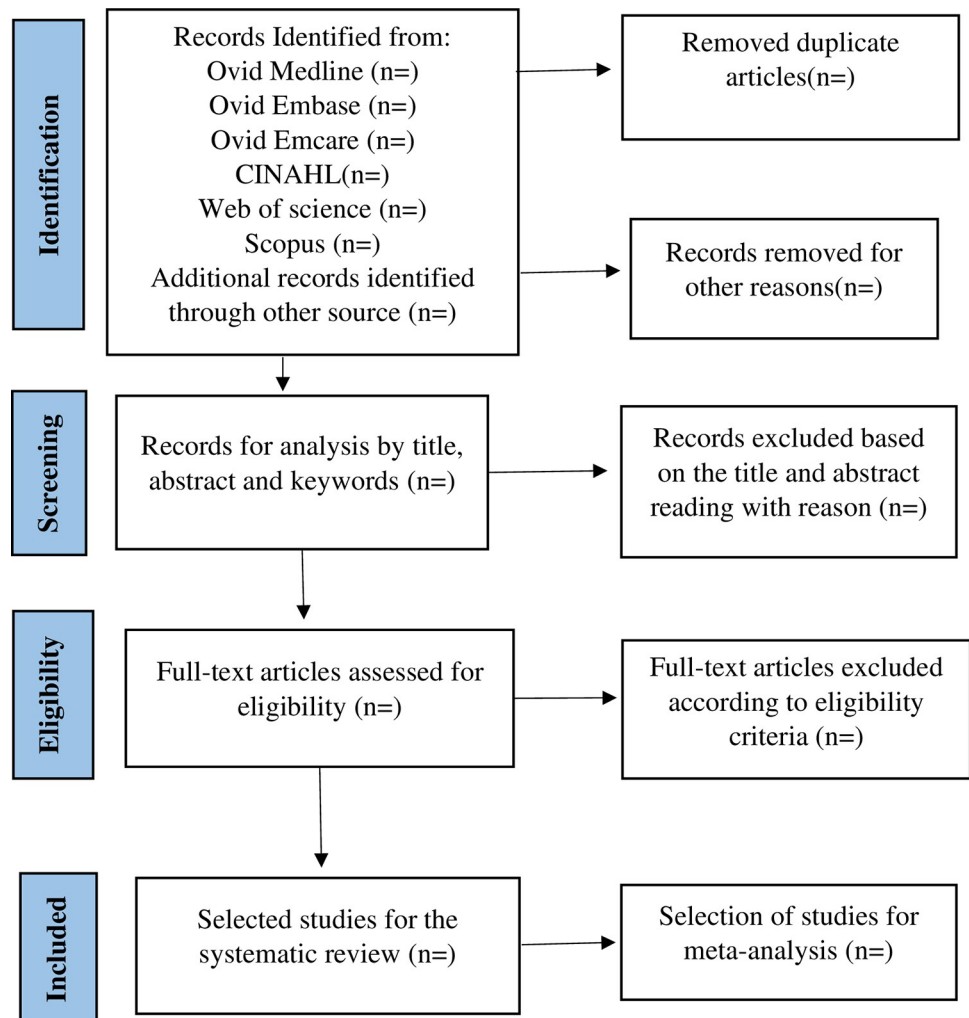

**Fig 1. Flowchart for the selection of articles based on PRISMA-P.** (n): is the number of articles that will be included at each stage.

Reasons for exclusion of studies reaching the full-text stage will be collated. We will use a PRISMA flowchart to report the results of the study screening and selection process (See Fig 1).

## Data extraction

Data will be extracted by two independent reviewers from all included studies using a standard data extraction tool. Fields to be extracted from eligible studies include (1) author name, (2) year of publication, (3) country of origin, (4) study design, (5) Study sample, (6) nature of exposure, (7) maternal trimester during pregnancy employment (exposure timing), (8) outcome, (9) confounders considered, (10) effect estimates, and (11) authors conclusion. For any missing data, we will contact the first or corresponding author to get additional information.

## Risk of bias (quality) assessment

Risk of bias for the included studies will be assessed using the critical assessment tools of the Joanna Briggs Institute (JBI) [39]. The tool consists of 8 items for analytical cross-sectional

studies, 11 for cohorts, 10 for case-control studies, and 13 items for randomized control trials. Responses to each item are coded as "yes", "no", "unknown", "not applicable".

The methodological quality of prospective cohort, case-control, and cross-sectional studies will be evaluated and screened for potential sources of bias, including inappropriate sampling, flawed measurement of exposure, flawed measurement of outcomes, incomplete outcomes, unidentified confounding factors, and inappropriate statistical analysis. There will also be consideration of additional sources of bias for particular study designs. For example, bias such as inadequate follow-up time and inappropriate follow-up strategies will be screened for prospective cohort studies, unclear inclusion criteria and study subjects will be screened for cross-sectional studies, and studies with incomparable cases and controls will be screened for case-control studies. A study will be deemed to have a high risk of bias if it has more than 70% of "yes" responses, a moderate risk of bias between 50%-69%, and a low risk of bias < 50%.

The quality assessment will be conducted independently by two reviewers. If there is a disagreement, a third reviewer will be invited to participate in quality assessment, and a decision will be made through discussion.

### Evidence synthesis

The quality of evidence across each exposure and the outcome of interest will be examined by using the GRADE (Grade of Recommendations, Assessment, Development, Evaluation) method [40]. The quality of the evidence will be rated as high, moderate, low, or very low. RCT evidence is initially rated as having a "high" degree of certainty, but this assessment may be downgraded if risk of bias, indirectness, inconsistency, imprecision and publication bias are felt to exist. Evidence of all observational studies will be given a "low" degree of certainty rating, but this assessment may be upgraded when there is evidence for a large magnitude of effect, dose-response, counteracting plausible residual bias, or confounding, the initial "low" rating will be upgraded.

This review will use the GRADE approach to develop practical messages [41, 42]. Accordingly, a high level of evidence will lead to "strong recommendations". A moderate level of evidence will lead to "practice considerations". For all evidence levels below moderate, the message will be "not enough evidence from the scientific literature to guide policymakers, clinicians, and patients.

If one of the three domains that can increase certainty in a body of evidence (usually from non-randomized studies) is included, consider rating up the grade of certainty, especially if it is noted in most studies. On the other hand, if three of the five domains that can decrease certainty in a body of evidence (typically from non-randomized studies) is noted, consider rating down the grade of certainty, particularly if it is noted in the majority of studies (See Table 2: summary of finding table which will be used for grading).

**Table 2. GRADE summary of findings: The relationship between physical occupational risks and preterm birth.**

| Quality assessment | | | | | | | | | |
|---|---|---|---|---|---|---|---|---|---|
| No of studies | Study design | Limitations | Inconsistency | Indirectness of evidence | Imprecision | Publication bias | Effect | Certainty of evidence | Strength of message |
| | | | | | | | | | |
| | | | | | | | | | |
| | | | | | | | | | |
| | | | | | | | | | |
| | | | | | | | | | |
| | | | | | | | | | |
| | | | | | | | | | |
| | | | | | | | | | |

Certainty of evidence rating will be done by two independent reviewers. If there is disagreement between the two reviewers at rating, for recommendation and implication a discussion will be held between the two reviewers to determine if consensus can be reached. If consensus cannot be reached, a third researcher will be consulted to resolve a difference for evidence rating.

## Meta- analysis

Meta-analyses will only be performed if there are sufficient studies having a similar definition of exposure and outcomes of interest. We will use the generic inverse variance method to apply meta-analysis with random effects modelling to investigate the association between maternal physical occupational risks and preterm birth. We will determine a pooled odds ratio from all studies that gave an adjusted odds ratio (OR) or risk ratio (RR) with a 95% confidence interval (CI) for the outcome of interest (preterm birth). As a measure of the heterogeneity between studies, visual inspection of forest plots and $I^2$ statistics tests will be used. Publication bias will be investigated using the Egger's weighted regression test and the Begg's test. In cases where meta-analysis is not possible, we will synthesize the data narratively. When multiple publications come from the same sample or data source, we will use the one that is the most comprehensive. The meta-analysis will be done by using Stata software. In cases where meta-analysis is possible subgroup analysis will be performed by study design and exposure type.

## Ethics and dissemination

Ethical approval will not be required due to the absence of primary data collection. The findings of this systematic review may be presented at national and international conferences and published in a peer-reviewed journal.

## Updates to study protocol

If any changes to the review protocol are required, these changes will be mentioned and enlisted as supplementary information along with a final manuscript and updated on the PROSPERO register.

# Discussion

Babies born prematurely (i.e., before 37 weeks of pregnancy) may have more health problems at birth and later in life than babies born between the 37th and 42nd week of pregnancy [43]. Premature birth is the leading cause of neonatal death and can have short- and long- term effects such as longer hospital stays, being readmitted to the hospital, respiratory distress, cerebral palsy, mental retardation, visual hearing impairments, poor health and growth, and chronic respiratory, cardiac, renal, endocrine system disorders later in life [18–21, 43].

Working pregnant women may be exposed to various occupational hazards during the course of their pregnancy [35, 44]. Occupational risks such as physical, chemical, biological and psychosocial exposures could increase the chance of adverse maternal and neonatal outcome including preterm birth [7, 18, 22, 24, 32]. Collecting and synthesising evidence can be a step towards a better understanding of the effect of maternal physical occupational predictors of preterm birth.

This review will synthesise current evidence to identify and understand physical occupational risks associated with preterm birth. This systematic review will explore, compile, and critically review the evidence on the effects of maternal physical occupational risk on preterm birth. Currently, there are inconclusive review findings on the effect of maternal physical

occupational risk on preterm birth. This review has a limitation that may exclude non-English language articles and thus may increase the risk of bias. However, to keep the transparency of this review we intend to exclude non-English language during the eligibility assessment stage. We will make recommendations for the obstetric community, occupational health services, and policy makers to promote the health of pregnant women.

## Supporting information

**S1 File. Preferred Reporting Items for Systematic reviews and Meta-Analysis (PRISMA) checklist.**
(DOCX)

**S2 File. Draft of search strategy or minimal data set.**
(DOCX)

## Acknowledgments

We are very grateful for the unending assistance of Alfred Ian Potter research and training librarian, Lorena Romero.

## Author Contributions

**Conceptualization:** Haimanot Abebe Adane, Ross Iles, Alex Collie.

**Investigation:** Ross Iles, Alex Collie.

**Methodology:** Haimanot Abebe Adane, Ross Iles, Jacqueline A. Boyle, Alex Collie.

**Resources:** Alex Collie.

**Software:** Haimanot Abebe Adane, Ross Iles, Alex Collie.

**Supervision:** Ross Iles, Jacqueline A. Boyle, Alex Collie.

**Validation:** Haimanot Abebe Adane, Ross Iles, Jacqueline A. Boyle, Alex Collie.

**Visualization:** Haimanot Abebe Adane, Ross Iles, Jacqueline A. Boyle, Alex Collie.

**Writing – original draft:** Haimanot Abebe Adane, Ross Iles, Alex Collie.

**Writing – review & editing:** Haimanot Abebe Adane, Ross Iles, Jacqueline A. Boyle, Alex Collie.

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
