## [Decision Letter · Decision Letter 0]

26 Jan 2023

PONE-D-22-29810Maternal occupational risk factors and Preterm birth: Protocol for a systematic review and meta-analysisPLOS ONE

Dear Dr. Adane,

Thank you for submitting your manuscript to PLOS ONE. After careful consideration, we feel that it has merit but does not fully meet PLOS ONE’s publication criteria as it currently stands. Therefore, we invite you to submit a revised version of the manuscript that addresses the points raised during the review process.

We look forward to receiving your revised manuscript.

Kind regards,

Ricardo Ney Oliveira Cobucci, Ph.D

Academic Editor

PLOS ONE

Journal Requirements:

Additional Editor Comments :

The authors prepared a systematic review protocol with the aim of assessing whether maternal occupational risk factors increase the risk of preterm delivery. Despite having previously published systematic reviews, the results are not conclusive and are still contradictory, which justifies the proposal of a new protocol.

However, as you can see, the reviewers have requested revisions to your manuscript. We are certainly willing to reconsider a revised submission, but please know that this is not preliminary acceptance of your paper. When returning your revised manuscript, please be sure to include a point-by-point summary of the suggestions of the reviewers that specifies how and where in the text you have addressed the suggestions.

In addition, consider reviewing the inclusion of manuscripts exclusively published in English in the protocol, as today with tools such as Google Translate and others, language limitations in the search strategy for a systematic review are not justified, as they facilitate the translation and understanding of the content of articles published in other languages. A search only for manuscripts published in English causes an important bias that limits the external validity of the systematic review, as well as compromising its results.

Finally, why did you decide to include only manuscripts published after the year 2000 in your search strategy? Are there no studies relating maternal occupational risk factors and preterm birth published before 2000? If they exist, the authors need to review this criterion. Another recommendation is that the authors add keywords related to the types of study in table 1, such as prospective cohort, retrospective cohort, case control and cross-sectional studies.

Reviewers' comments:

Reviewer's Responses to Questions

**Comments to the Author**

1. Does the manuscript provide a valid rationale for the proposed study, with clearly identified and justified research questions?

Reviewer #1: Yes

Reviewer #2: Yes

2. Is the protocol technically sound and planned in a manner that will lead to a meaningful outcome and allow testing the stated hypotheses?

Reviewer #1: Yes

Reviewer #2: Yes

3. Is the methodology feasible and described in sufficient detail to allow the work to be replicable?

Reviewer #1: Yes

Reviewer #2: Yes

4. Have the authors described where all data underlying the findings will be made available when the study is complete?

Reviewer #1: Yes

Reviewer #2: Yes

5. Is the manuscript presented in an intelligible fashion and written in standard English?

Reviewer #1: Yes

Reviewer #2: Yes

6. Review Comments to the Author

You may also provide optional suggestions and comments to authors that they might find helpful in planning their study.

Reviewer #1: The manuscript seems to be well-written. I have some minor comments.

1. (Introduction) I think that systematic review of this topic is important. In contrast, could you explain in Introduction why a protocol for the systematic review needs to be published? I think that the authors should submit a manuscript after conducting the research.

2. (Introduction) Weren’t there any systematic reviews that investigated an association between preterm birth and occupational physical risk in the past?

3. (Meta-analysis section in Methods) It is written hat “a pooled odds ratio from ~ risk ratio”, how do you obtain a pooled odds ratio from risk ratio? Using risk ratio as a odds ratio for some studies might not be a good idea.

Reviewer #2: This manuscript by Adane et al. is useful for public health reading especially in the area of maternal and foetal health.

Minor comments

Consistency with "preterm" or "pre-term"

Figure 1 has (n=) with no explanation for its meaning

Inclusion criteria: Both singleton and twin pregnancies, nulliparous and multiparous included? Please clarify

7. PLOS authors have the option to publish the peer review history of their article (what does this mean?). If published, this will include your full peer review and any attached files.

Reviewer #1: No

Reviewer #2: **Yes: **ENOCH ODAME ANTO

---

## [Author Response · Author response to Decision Letter 0]

18 Feb 2023

Authors response

Maternal occupational risk factors and Preterm birth: Protocol for a systematic review and meta-analysis 

Corresponding Author; Haimanot Abebe Adane: haimanot.adane@monash.edu

Authors

Haimanot Abebe Adane: Haimanot.Adane@monash.edu

Ross Iles: ross.iles@monash.edu

Jacqueline A. Boyle: jacqueline.boyle@monash.edu

Alex Collie: Alex.Collie@monash.edu

Manuscript ID: PONE-D-22-29810

Journal: PLOS ONE

Article type: Study Protocol (Systematic review and meta-analysis)

To editors and reviewers

First of all, the authors would like to express their gratitude to the editors of the “PLOS ONE” journal editor and the respected reviewers for reviewing our manuscript and offering valuable suggestions to enhance its scientific merit. We have updated the manuscript with corrections in response to the comments made. As a result, all comments have been accepted and integrated into the revised manuscript. 

A point-by-point response to Editor 

Editor: Consider reviewing the inclusion of manuscripts exclusively published in English in the protocol, as today with tools such as Google Translate and others, language limitations in the search strategy for a systematic review are not justified, as they facilitate the translation and understanding of the content of articles published in other languages. A search only for manuscripts published in English causes an important bias that limits the external validity of the systematic review, as well as compromises its results

Authors: While the exclusion of non-English language articles may lead to bias and miss important articles, some scholars have argued that the exclusion of non-English articles has a limited impact on the finding and overall conclusion of the review (https://doi.org/10.1016/j.jclinepi.2019.10.011). From a practical perspective excluding non-English language articles during the search stage of a review risks the exclusion of relevant non-English language articles, where language values have been incorrectly defined or are missing. Excluding non-English language articles during the eligibility assessment stage instead allows the reason for ineligibility to be recorded, providing greater transparency about the number of articles excluded on this basis. We will exclude non-English language at the eligibility assessment stage due to the practical barriers such as the high cost and time commitment associated with translating articles. We have considered the limitations of translation software (e.g Google Translate): Based on our experience, and scholarly point of view google translate often produces translations that contain significant grammatical errors and do not have a system to correct translation errors [https://doi.org/10.1186/2046-4053-2-97]. This may lead to missing or misinterpreting key evidence, which may limit the generalisability of findings. 

Editor: Finally, why did you decide to include only manuscripts published after the year 2000 in your search strategy? Are there no studies relating maternal occupational risk factors and preterm birth published before 2000? If they exist, the authors need to review this criterion. 

Authors: The reason for deciding to include only articles published after the year 2000 is that, over the years, not only has the proportion of women in the workforce changed, but also the working conditions for pregnant women. Recently it is more common to have modifications of working conditions during pregnancy, paid maternity leave, or health benefits by law. To provide a contemporary answer to whether or not nowadays specific physical activities or working conditions exert an influence on preterm birth, we only seek recent studies for this systematic review using more recent studies will give a better reflection of today’s risk of preterm birth. By limiting studies to published in 2000 or more recent, we will be reviewing more than 20 years of research while ensuring the findings reflect contemporary working conditions.

Editor: Another recommendation is that the authors add keywords related to the types of study in table 1, such as prospective cohort, retrospective cohort, case-control and cross-sectional studies. 

Authors: The goal of systematic review searches is to identify all relevant studies on a topic. Systematic review searches are therefore typically quite extensive. However, it may be necessary to strike a balance between the sensitivity and precision of our search. Increasing the comprehensiveness of a search will reduce its precision and will retrieve more non-relevant results. Thus, considering your suggestion we follow the standard search strategy technique (PICO). 

A point-by-point response to (Reviewer # 1)

Reviewer 1: (Introduction) I think that a systematic review of this topic is important. In contrast, could you explain in the Introduction why a protocol for the systematic review needs to be published? I think that the authors should submit a manuscript after conducting the research.

Authors: The importance of publishing our protocol for the systematic review appears Line 111-114). We do intend to submit a manuscript detailing the review findings. 

Reviewer 1: (Introduction) Weren’t there any systematic reviews that investigated an association between preterm birth and occupational physical risk in the past?

Authors: While there are previous reviews exist, the reasons we are conducting this SLR and Meta-analysis are briefly described in Lines 93-100. In more detail: 

1- While the evidence from previous reviews is useful, their authors have reported conflicting or weak evidence and as such have concluded that it is challenging to provide explicit recommendations for clinical practice or policy. 

2- A number of prior reviews have not utilised rigorous methodological standards for reporting on study quality

3- None of the reviews examined the impacts of whole-body vibration on preterm birth, and nor have they sought to differentiate between medically indicated or spontaneous preterm birth

4- Further, the included evidence in most reviews reflects working conditions of the 1960’s and 2000’s. In many occupations and nation, working conditions have changed dramatically throughout the early 21st century and thus the nature, prevalence and impacts of occupational physical health risks has also changed. 

Reviewer 1: (Meta-analysis section in Methods) It is written hat “a pooled odds ratio from ~ risk ratio”, how do you obtain a pooled odds ratio from risk ratio? Using risk ratio as a odds ratio for some studies might not be a good idea.

Authors: We intend to transform risk ratios into odd ratios by using the formula; RR = OR / (1 – p + (p x OR)), where p is the risk in the control group. 

A point-by-point response to (Reviewer # 2)

Reviewer 2: This manuscript by Adane et al. is useful for public health reading, especially in the area of maternal and fetal health. Minor comments Consistency with "preterm" or "pre-term"

Authors: This is now consistent throughout the manuscript.

Reviewer 2: Figure 1 has (n=) with no explanation for its meaning

Authors: We have made corrections. (See; Line 174)

Reviewer 2: Inclusion criteria: Both singleton and twin pregnancies, nulliparous and multiparous included? Please clarify

Authors: We have clarified in the revised manuscript that we will include a singleton pregnancy and both nulliparous and multiparous women. (See; Line 125) Since over 60% of twin and nearly all higher-order multiples are premature (born before 37 weeks) and intended to control the confounding effect of multiple births we only include a singleton pregnancy.

---

## [Decision Letter · Decision Letter 1]

15 Mar 2023

Maternal occupational risk factors and Preterm birth: Protocol for a systematic review and meta-analysis

PONE-D-22-29810R1

Dear Dr. Adane,

We’re pleased to inform you that your manuscript has been judged scientifically suitable for publication and will be formally accepted for publication once it meets all outstanding technical requirements.

Kind regards,

Ricardo Ney Oliveira Cobucci, Ph.D

Academic Editor

PLOS ONE

Additional Editor Comments (optional):

Reviewers' comments:

Reviewer's Responses to Questions

**Comments to the Author**

1. Does the manuscript provide a valid rationale for the proposed study, with clearly identified and justified research questions?

Reviewer #1: Yes

Reviewer #2: Yes

2. Is the protocol technically sound and planned in a manner that will lead to a meaningful outcome and allow testing the stated hypotheses?

Reviewer #1: Yes

Reviewer #2: Yes

3. Is the methodology feasible and described in sufficient detail to allow the work to be replicable?

Reviewer #1: Yes

Reviewer #2: Yes

4. Have the authors described where all data underlying the findings will be made available when the study is complete?

Reviewer #1: Yes

Reviewer #2: No

5. Is the manuscript presented in an intelligible fashion and written in standard English?

Reviewer #1: Yes

Reviewer #2: Yes

6. Review Comments to the Author

You may also provide optional suggestions and comments to authors that they might find helpful in planning their study.

Reviewer #1: Thank you for the comments.

It is true that risk ratios can be transformed into odd ratios by using the formula; RR = OR / (1 – p + (p x OR)).

However, in case control studies, p cannot be obtained in general.

Therefore, RRs and ORs need to be analyzed separately, or RRs need to be transformed to ORs.

Reviewer #2: Authors have responded to all comments on the manuscript entitled, "Maternal occupational risk factors and Preterm birth: Protocol for a systematic review and meta-analysis"

7. PLOS authors have the option to publish the peer review history of their article (what does this mean?). If published, this will include your full peer review and any attached files.

Reviewer #1: No

Reviewer #2: No

---

## [Editor Report · Acceptance letter]

21 Mar 2023

PONE-D-22-29810R1 

Maternal occupational risk factors and preterm birth: protocol for a systematic review and meta-analysis 

Dear Dr. Adane:

I'm pleased to inform you that your manuscript has been deemed suitable for publication in PLOS ONE. Congratulations! Your manuscript is now with our production department. 

Kind regards, 

on behalf of

Dr. Ricardo Ney Oliveira Cobucci 

Academic Editor

PLOS ONE